# Performance of Salivary Extracellular RNA Biomarker Panels for Gastric Cancer Differs between Distinct Populations

**DOI:** 10.3390/cancers14153632

**Published:** 2022-07-26

**Authors:** Karolina Elżbieta Kaczor-Urbanowicz, Mustafa Saad, Tristan R. Grogan, Feng Li, You Jeong Heo, David Elashoff, Robert S. Bresalier, David T. W. Wong, Yong Kim

**Affiliations:** 1Center for Oral and Head/Neck Oncology Research, School of Dentistry, University of California at Los Angeles, Los Angeles, CA 90095, USA; kkaczor@ucla.edu (K.E.K.-U.); mzsaad@g.ucla.edu (M.S.); fengli@ucla.edu (F.L.); heo893@naver.com (Y.J.H.); 2UCLA Institute for Quantitative and Computational Biosciences, University of California at Los Angeles, Los Angeles, CA 90095, USA; 3UCLA Section of Orthodontics, University of California at Los Angeles, Los Angeles, CA 90095, USA; 4Section of Biosystems and Function, School of Dentistry, University of California at Los Angeles, Los Angeles, CA 90095, USA; 5Department of Medicine Statistics Core, David Geffen School of Medicine at the University of California at Los Angeles, Los Angeles, CA 90024, USA; tgrogan@mednet.ucla.edu (T.R.G.); DElashoff@mednet.ucla.edu (D.E.); 6Department of Gastroenterology, Hepatology and Nutrition, Division of Internal Medicine, The University of Texas MD Anderson Cancer Center, Houston, TX 77030, USA; 7UCLA’s Jonsson Comprehensive Cancer Center, Los Angeles, CA 90024, USA

**Keywords:** biomarkers, validation, gastric cancer, exRNA, saliva

## Abstract

**Simple Summary:**

Gastric cancer (GC) is the fourth most common cancer that occurs worldwide, affecting specifically the Asian population. Currently, there are no available screening programs for GC in United States. Since saliva is a highly desirable body fluid for developing biomarkers of cancer screening, early detection, and monitoring, we previously reported that salivary extracellular RNAs could be developed to detect gastric cancer in a Korean cohort, and here, we validate them in a U.S. cohort. Our study emphasizes the importance of population-specific biomarker development and validation, and specifically, the noninvasive nature of salivary biomarkers for population-based screening in at-risk populations.

**Abstract:**

Gastric cancer (GC) has the fifth highest incidence among cancers and is the fourth leading cause of cancer-related death GC has predominantly a higher number of cases in certain ethnic groups such as the Korean population. GC found at an early stage is more treatable and has a higher survival rate as compared with GC found at a late stage. However, a diagnosis of GC is often delayed due to the lack of early symptoms and available screening programs in United States. Extracellular RNA (exRNA) is an emerging paradigm; exRNAs have the potential to serve as biomarkers in panels aimed at early detection of cancer. We previously reported the successful use of a panel of salivary exRNA for detecting GC in a high-prevalence Korean cohort, and that genetic changes reflected cancer-associated salivary exRNA changes. The current study is a case-control study of salivary exRNA biomarkers for detecting GC in an ethnically distinct U.S. cohort. A model constructed for the U.S. cohort combined demographic characteristics and salivary miRNA and mRNA biomarkers for GC and yielded an area under the receiver operating characteristic (ROC) curve (AUC) of 0.78. However, the constituents of this model differed from that constructed for the Korean cohort, thus, emphasizing the importance of population-specific biomarker development and validation.

## 1. Introduction

Gastric cancer (GC) is an aggressive type of cancer that remains a healthcare burden worldwide [1]. For 2022, the American Cancer Society has estimated that there will be about 26,380 new cases of gastric cancer in the USA and about 11,090 deaths (https://seer.cancer.gov/statfacts/html/stomach.html (accessed on 10 May 2022)). Gastric cancer accounts for ~1.5% of all new cases of cancers in the USA each year. The incidence of GC in the United States has relatively decreased, however, in Western countries, approximately half of patients present with locally advanced or metastatic GC at diagnosis, and an additional 40% to 60% of those patients undergoing resection of gastric adenocarcinoma relapse after surgery [2]. Thus, early detection of this type of cancer is the main goal to reduce mortality, as the 5-year survival rate of early detected cases can reach >95% [3].

Many studies from East Asian countries have shown that screening methods, especially endoscopic screening for detecting early-stage GC, have resulted in a reduction in mortality. However, population-based screening programs do not exist in the USA, because of the low incidence of GC overall [4].

Although upper gastroesophageal endoscopy with targeted and random biopsies remains to be the gold standard for the detection of GC, other screening tools have been implemented in high-risk countries such as pepsinogens (PGs) including PG 1 and PG 2, gastrin-17, and *Helicobacter pylori* (*H. pylori*) IgG antibody tests. Blood-based tumor markers such as carcinoembryonic antigen (CEA) and carbohydrate antigen 19-9 (CA19-9) have also been used for the detection of GC, but have low sensitivity and specificity for early-stage disease [5,6].

Screening programs for GC vary among countries, depending on prevalence and cost-effectiveness [6]. We previously reported the use of extracellular RNA (exRNA) biomarkers in saliva as a diagnostic tool for screening and/or risk assessment for GC [7]. In a study of a Korean cohort of subjects, we identified 30 mRNA and 12 miRNA biomarkers that were associated with the expression pattern and presence of GC. A configured biomarker panel consisted of three mRNAs (*SPINK7*, *PPL*, and *SEMA4B*) and two miRNAs (*miR-140-5p* and *miR-301a-3p*) that were all significantly downregulated in the GC group, and yielded an area under the receiver operating characteristic (ROC) curve (AUC) of 0.81 (95% CI 0.72–0.89). When combined with demographic factors, the AUC of the biomarker panel reached 0.87 (95% CI 0.80–0.93) in differentiating subjects with GC from those without cancer. Since a lower expression of these salivary markers was indicative of GC, a comprehensive cut-off validation study would be necessary to develop these markers for the screening of GC in the general population.

It is known that the pathogenesis of GC depends on multiple etiological factors and ethnicity could obviously be one of determining factors [8]. However, it is also possible that there are common biological alterations that may contribute to the pathogenesis of disease and these genetic changes may be cancer-associated salivary exRNA alterations. The miRNAs and exRNAs are endogenous, small non-coding RNA molecules that post-transcriptionally modulate gene expression [9]. Because these molecules are stable in different body fluids including saliva, analysis of these molecules can lead to an important, noninvasive diagnostic and prognostic tool for GC screening. However, the expression of biomarkers may differ based on the population investigated. A key requirement of biomarker validation for clinical and regulatory purposes is that of intended use. A biomarker or panel of biomarkers that may differentiate disease from normal in one population may not perform similarly in a different population. While we have previously demonstrated that a particular panel of salivary biomarkers may differentiate subjects with GC from those without cancer in an Asian population with a high prevalence of gastric cancer, it is unclear whether this same panel of markers would perform similarly in a low-prevalence U.S. population. The current study represents a case-control study of salivary exRNA biomarkers in a U.S. cohort.

## 2. Materials and Methods

### 2.1. Saliva Collection and Processing

Unstimulated whole saliva was collected from 50 newly diagnosed treatment-naive patients with histologically proven GC (stages I–IV) and 50 control subjects without GC, based on recent endoscopic results at the University of Texas MD Anderson Cancer Center, USA, using standard operating procedures (SOPs) developed for our prior study of a Korean cohort [7,10]. Subjects were asked to avoid oral hygiene measures, eating, drinking, and gum chewing at least 1 h prior to saliva collection. The subjects rinsed with tap water (10 mL) for 30 s about 10 min prior to saliva collection and expectorated. Clinical samples were collected in sterile tubes, lasting 5–10 min per collection (at least 5 mL of saliva), and kept on ice through the entire process. All samples were processed, approximately 1 h after collection. First, samples were centrifuged in a refrigerated centrifuge at 2400× *g* for 15 min at 4 °C, and the supernatant was processed immediately for the concurrent stabilization of proteins and RNA by the inclusion of a protease inhibitor cocktail (aprotinin, 3-phenylmethylsulfonyl fluoride (3-PMSF) (Sigma-Aldrich, St. Louis, MO, USA), sodium orthovanadate (Na3VO4) (Sigma-Aldrich, St. Louis, MO, USA)) and RNase inhibitor (Invitrogen SUPERase·In RNase Inhibitor, Thermo Fisher Scientific, Austin, TX, USA) based on our saliva standard operating procedure (SOP) [11]. These samples were aliquoted into smaller cryo-vials, labeled, and frozen at −80 °C.

This study, including the patient consent process, was approved by the Institutional Review Board for Human Studies at the University of Texas MD Anderson Cancer Center (IRB number PA17-0583). The control group consisted of subjects undergoing upper endoscopy for dyspepsia or gastroesophageal reflux-like symptoms and documented to have no neoplasia. Patient-level clinical demographics were obtained (age, gender, ethnicity, smoking history, staging, and diagnosis). The study was performed from 19 October 2017 to 13 June 2019.

### 2.2. RNA Isolation from Saliva Samples

Total RNA from 50 blinded GC subjects and 50 non-GC control subjects was isolated by using a Qiagen miRNeasy Micro kit (Qiagen, Germantown, MD, USA). The 250 μL samples of cell-free saliva was used to isolate total RNA using a modified protocol successfully used in the lab for isolating salivary RNA [10]. The final RNA was eluted in 14 μL of water.

### 2.3. Validation of miRNA GC Markers

The biomarker panel used in this study contained two miRNAs (*miR-140-5p* and *miR-301a-3p*); *U6* snRNA and *miR-197* were used as the reference genes. TaqMan miRNA assays (Thermo Fisher Scientific, Austin, TX, USA), containing these four small RNA genes, were ordered from Applied Biosystems (Foster City, CA, USA). The protocol was similar to that recommended by the manufacturer for creating custom reverse transcription (RT) and preamplification primer pools using TaqMan MicroRNA Assays (Thermo Fisher Scientific, Austin, TX, USA). Total RNA (3 ng) was converted to cDNA using a TaqMan MicroRNA Reverse Transcription Kit (Applied Biosystems, Foster City, CA, USA). After RT, the product was preamplified using SsoAdvance PreAmp Supermix (Bio-Rad, Hercules, CA, USA) and preamplification primer pool. The preamplified product was diluted 2 times prior to miRNA quantification. The qPCR reactions for each candidate miRNA were performed in triplicate on a Roche LightCycler 480 II (Roche, San Francisco, CA, USA). The average threshold cycle (Cq) was examined and *U6* snRNA and *miR-197* were used as the reference genes for normalizing the data.

### 2.4. Validation of mRNA GC Markers

Three selected candidate mRNA biomarkers (3 mRNAs (*PPL*, *SEMA4B,* and *SPINK7)* as well as 2 reference genes (*GAPDH* and *ACTB*)) generated by microarray profiling were validated by nested real-time quantitative polymerase chain reaction (RT-qPCR) (RT-PCR followed by a separate SYBR green quantitative polymerase chain reaction (qPCR)) on the new set of samples from MD Anderson Cancer Center (blinded 50 GC and 50 non-GC). The gene accession numbers and primer sequences used for the transcriptomic biomarker validation are shown in Appendix A. The qPCR primers were designed using the Primer3 software and synthesized by Sigma-Genosys after performing a Primer-BLAST search. The primer sequences were designed to avoid any known single-nucleotide polymorphism region in the target gene. All the amplicons were intron spanning. The RT-qPCR assay followed the Minimum Information for Publication of Quantitative Real-Time PCR Experiment guidelines and was performed in duplicate with each biomarker candidate. The specificity of the PCR product for each gene was confirmed with melting curve analysis and 3% agarose gel analysis.

### 2.5. RT-qPCR Preamplification for Validation of mRNA Candidates

The multiplex RT-PCR preamplification was performed with an Invitrogen SuperScript III Platinum One-Step qRT-PCR System (Thermo Fisher Scientific, Austin, TX, USA) with a pool of outer primers at 100 nM each. The reaction mixture was prepared on ice, and then loaded into the preheated thermocycler. The amplification was performed as follows: 2 min at 60 °C; 30 min at 50 °C; 2 min at 95 °C; and 15 cycles of 15 s at 95 °C, 30 s at 50 °C, 10 s at 60 °C, and 10 s at 72 °C; with a final extension for 10 min at 72 °C and cooling to 4 °C. Immediately after RT-qPCR, 10 μL of the reaction was treated with 4 μL of Exo-SAP-IT (Thermo Fisher Scientific, Austin, TX, USA) for 15 min at 37 °C to remove excess primers and deoxynucleotide triphosphates (dNTPs), and then heated to 80 °C for 15 min to inactivate the enzyme mix. The preamplified complementary DNAs (cDNAs) were then diluted by adding water to 200 μL (20-fold) to enable the qPCR of all targets.

### 2.6. qPCR for Validation of mRNA Candidates

Singleplex qPCR was performed in 10 µL reactions with 2 µL of each preamplified cDNA sample and the inner primers at 200 nM each. The reaction was conducted with a SYBR Green I Master mix in LightCycler 480 (Roche Diagnostics, Indianapolis-Marion County, Indiana) instrument. After 10 min of polymerase activation at 95 °C, 40 cycles of 15 s at 95 °C and 60 s at 60 °C were performed, followed by melting curve analysis. Three controls including one RT control, no-template control, and positive control with universal human RNA were performed with every candidate on each sample.

### 2.7. Statistical Analysis for qPCR

The qPCR analyses were all done in triplicate. For the miRNA analysis, data were analyzed using the RQ Manager software version 1.2 and DataAssist software version 3.0 (Applied Biosystems). Similarly, the ∆Cq value was computed using RNA polymerase III transcribed *U6* small nuclear RNA as the reference gene [7]. For the mRNA analysis, the ∆Cq of each biomarker candidate was calculated by subtracting the Cq value of the housekeeping genes (*GAPDH* and *ACTB*) from the raw Cq value in the same sample. ∆Cq values for mRNA and miRNA were compared between groups using the Wilcoxon rank-sum test.

## 3. Results

### 3.1. Clinicopathological Characteristics of Patients

The patients’ characteristics and study variables are summarized between groups (GC vs. control) using mean (SD) and frequency (%) and compared between groups using the two-sample *t*-test or chi-square test (Table 1).

### 3.2. miRNA RT-qPCR

Next, we constructed a model with the demographic terms (age, gender, and smoking history) plus the two candidate miRNA markers for GC (computing the dCT by subtracting the reference gene (*U6*) from our candidate markers (*miR-140* and *miR-301a*)). This was the same reference gene (*U6*) used in our prior study [7]. In this study, we found the CT values for *U6* were 15.37 ± 2.64 in the non-GC control group and 15.53 ± 2.70 in the GC patient group (*p* = 0.809 by *t*-test). The *p*-values show no significant differences between GC patients and non-GC controls, suggesting *U6* is a good reference gene for salivary exRNA quantification. To reduce the potential bias from one reference gene, we also tested *miR-197* as an extra reference small RNA. We found the CT values for *miR-197* were 16.20 ± 1.82 in the non-GC control group and 16.19 ± 1.85 in the GC patient group (*p* = 0.796). Next, we compared the AUC between the model with only demographic factors to the model utilizing demographic and miRNA data using the DeLong’s test (Table 2). Analyses were conducted using IBM SPSS V25 (Armonk, NY, USA) and R V 3.6.1 (www.r-project.org (accessed on 20 March 2021), Vienna, AU, USA) and *p*-values < 0.05 were considered to be statistically significant. The AUC (95% CI) was 0.75 (0.65–0.84) for the GC group versus the non-GC group based on these two miRNA markers together with demographic factors. The markers (dCT) were both significant in that model (*miR-140* (*p* = 0.003), miR-301a (*p* = 0.002)). Interestingly, the demographic model alone yielded an AUC of only 0.68, while the combined model (demographic data with miRNA biomarkers) resulted in an improved AUC of 0.75 (DeLong *p*-value = 0.129).

Next, logistic regression models for GC status were constructed using demographic factors from our previous publication (age, gender, and smoking history) with the AUC (95% CI) and odds ratios (ORs) estimated (Table 3). Interestingly, the demographic factors in this U.S. cohort showed an AUC of 0.68 (95% CI 0.57–0.78), which was similar to the AUC of 0.69 (95% CI 0.59–0.79) in the Korean cohort that we previously reported [7].

### 3.3. mRNA RT-qPCR

We constructed a new model for the U.S. cohort using the same variables in three different ways, as reported in our previous report based on a Korean cohort [7]:(1)Model 1, a new model with only demographic characteristics (AUC = 0.68, sensitivity = 62.7%, and specificity = 70.8%);(2)Model 2, a new model with demographic characteristics and miRNA biomarkers for GC (AUC = 0.75, sensitivity = 62.7%, and specificity = 81.3%);(3)Model 3, a new model with demographic characteristics, miRNA, and mRNA biomarkers for GC (AUC = 0.78, sensitivity = 62.7%, and specificity = 83.3%).

When mRNAs were combined with miRNA biomarkers and demographic features, the new combined model yielded the best AUC of 0.78 for differentiating subjects with GC from those without GC, with the highest specificity (sensitivity = 62.7% and specificity = 83.3%) (Figure 1) as compared with the models composed of only demographic features (AUC = 0.68, sensitivity = 62.7%, and specificity = 70.8%) or demographic features together with miRNAs (AUC = 0.75, sensitivity = 62.7%, and specificity = 81.3%). There was also a significant difference between Model 1 (AUC = 0.68) and Model 3 (AUC = 0.78) (Delong’s test *p*-value = 0.037), suggesting that the model with both miRNA and mRNA biomarkers for GC combined with demographic characteristics (Model 3) performed better than the model with only demographic characteristics (Model 1) with increased specificity (83.3% for Model 3 as compared with 70.8% for Model 1) (Figure 1, Table 4).

However, when we applied the coefficients as estimated from the prior Korean cohort study [7], the AUC was only 0.52 because of differences in the significance of individual demographic features as well as in performance of GC miRNA and mRNA biomarkers in the current U.S. cohort as compared with the Korean cohort.

Additionally, we also assessed the panel performance (demographic characteristics (demo) + miRNAs + mRNAs) in two separate scenarios defined as controls vs. early-stage GC (I, II) as well as controls vs. late-stage GC (III, IV). It appeared that the discrimination (AUC) of the control vs. early-stage GC model was 0.85 (0.72–0.99), whereas for the control vs. late-stage GC, the performance was 0.75 (0.64–0.85). Therefore, our panel may perform better in discriminating controls from early-stage GC, although this would need to be confirmed in a follow-up study (Table 3). 

## 4. Discussion

A growing number of studies have demonstrated the utility of exRNA as a reliable noninvasive approach for diagnosis, therapy, and prognosis of cancers [12]. Extracellular RNAs have been explored as biomarkers in a number of different biofluids and types of cancer, which include esophageal squamous cell carcinoma (ESCC) [13], lung cancer [14], brain cancers [15,16,17,18], prostate cancer [19], pancreatic cancer [20], colon cancer [21], and gastric cancer [22]. As of 2020, 45 clinical trials, in the USA and numerous other countries, have been reported that have focused on the use of exRNA and exosomes as clinical biomarkers of cancer [12]. These clinical trials have explored exRNAs as clinical biomarkers of various cancer types including lung and prostate cancers. Blood is a primary source of exRNAs that have been tested, but studies have also investigated urine. Especially saliva is being explored as an emerging biofluid that is easy to collect, and has been shown to reflect the spectrum of health and disease states found using serum [23,24].

Standards for validation of biomarkers require that they be applied in the population for which they are intended to be used [25]. A biomarker or panel of biomarkers which can differentiate disease from normal in one population may not perform similarly in a different population. We previously reported on a panel of salivary biomarkers which, when combined with specific demographic factors, differentiated subjects with GC from those without cancer in an Asian population with a high prevalence of gastric cancer. Our aim was to evaluate the performance of salivary exRNA biomarkers for GC, which we previously discovered and validated in Korean GC patients [7], in a U.S. population. Previously, 12 mRNA and 6 miRNA candidates were verified with a discovery Korean cohort by RT-qPCR and further validated with an independent Korean cohort (*n* = 200). The configured biomarker panel consisted of three mRNAs (*SPINK7*, *PPL*, and *SEMA4B*) and two miRNAs (*miR-140-5p* and *miR-301a*), which were all significantly downregulated in the GC group, and yielded an AUC of 0.81 (95% CI 0.72–0.89). When combined with demographic factors, the AUC of the biomarker panel reached 0.87 (95% CI 0.80–0.93) [7]. In our prior study, demographic characteristics (including age, gender, and smoking) were all highly significant predictors of case status [7], while, in the current U.S. MD Anderson Cancer Center cohort, only gender was significant (Table 4). However, this U.S. cohort (MD Anderson Cancer Center cohort) had more ethnic diversity including Caucasian (51% of the GC group and 65.3% of the control group), Black non-Hispanic (19.6% of the GC group and 12.2% of the control group), and Hispanic (17.6% of the GC group and 16.3% of the control group) subjects, with the Asian population as the least prevalent group (11.8% of theGC group and 6.1% of the control group). In a previous study [7], Asians constituted 100% of the GC and control groups for both mRNA and miRNA discovery and validation phases (Figure 2). In addition, in the current study, the samples were obtained from older patients (GC group, 61 years old and control group, 60 years old), fewer smokers (present or prior smoking, 43.1% of the GC group and 42.9% of the control group), and fewer males (62.7% of the GC group and 32.7% of the control group) as compared with the Korean cohort (Figure 2) [7].

It is unclear how these factors may account for differences between the two distinct populations with respect to biomarker profiles. Interestingly, the prevalence of *H. pylori* positivity in the U.S. population with gastric cancer was relatively low as compared with Asian populations with gastric cancer where *H. pylori* was a major risk factor. There was no difference, however, in the prevalence of *H. pylori* between cancer patients and controls in the U.S. cohort, but the number of subjects studied was small.

These demographic differences may be important factors to consider for validation of salivary exRNA GC biomarkers in two entirely independent patient cohorts (Korea vs. USA). Our study indicated that, overall, the demographic factors in this U.S. cohort were similar (AUC of 0.68) to those of the Korean cohort (AUC of 0.69) [7]. In our previous study with Korean subjects, we found a difference in nearly all the selected mRNAs (*ANXA1, CD24, CSTB, ERO1A, KRT4, KRT6A, PPL, RANBP9, S100A10, SEMA4B,* and *SPINK7*) and miRNAs for GC (*miR-140-5p*, *miR-374a, miR-454, miR-15b, miR-28-5p,* and *miR-301a*). They were all statistically significant (FDR-adjusted *p*-value <0.05). However, in the U.S. cohort, none of the miRNAs (*miR-140* and *miR-301*) and mRNAs (*SPINK7*, *PPL*, and *SEMA4B*) performed similarly (Appendix A). Therefore, any model constructed from the prior Korean cohort could not be generalized to the current U.S. cohort [7].

Interestingly, there was a statistically significant difference between a new model with only demographic characteristics (AUC = 0.68) and a new model with demographic characteristics, and miRNA/mRNA biomarkers for GC (AUC = 0.78) (Figure 1) (Delong’s test *p*-value = 0.037), indicating that the model with both miRNAs and mRNAs together with demographic characteristics (Model 3) was much better than the model with demographic characteristics alone (Model 1). Interestingly, our panel has a potential to perform better in discriminating non-GC controls from early-stage GC (I, II) (AUC = 0.85 (95% CI 0.72–0.99)) as compared with the late-stage GC (III, IV) (AUC = 0.75 (95% CI 0.64–0.85)), although this still would need to be confirmed in a follow-up study. Thus, we were able to validate a panel of salivary exRNA biomarkers with credible clinical performance for the detection of GC in a U.S. population. Our study confirms, again, the potential utility of salivary exRNA biomarkers in screening and risk assessment for GC.

### 4.1. Current Biomarker Performance

Most of the currently available published reports are based on blood-based biomarkers for GC. So et al. developed a 12-miR assay from serum specimens (*miR-140, miR-183, miR-30e, miR-103a, miR-126, miR-93, miR-142, miR-21, miR-29c, miR-424, miR-181a,* and *miR-340*) in a three-phase, multicenter study comprising 5248 subjects from Singapore and Korea in retrospective cohorts of 682 subjects [26]. Interestingly, one of the miRNAs was *miR-140*, the same as that investigated in our studies. The 12- miRNA panel yielded an AUC of 0.93 (95% CI 0.90–0.95) and an AUC of 0.92 (95% CI 0.88–0.96) in the discovery and verification cohorts, respectively. In this prospective study, overall sensitivity was 87.0% (95% CI 79.4–92.5%) at specificity of 68.4% (95% CI 67.0–69.8%). Interestingly, the AUC was 0.848 (95% CI 0.81–0.88), much higher than other more frequently used gastric tumor markers such as *H. pylori* serology (0.635), pepsinogen (PG) 1/2 ratio (0.641), PG index (0.576), ABC method (0.647), CEA (0.576), and CA19-9 (0.595) [26]. Other plasma miRNAs yielded AUCs from 0.65 to 0.75 for *miR-185, miR-20a, miR-210, miR-25*, and *miR-92b* [27]. In another study, *miR-181a-1* and *KAT2B* mRNA were identified as a combined predictor for GC with AUC > 0.95 [28], while the expression levels of *HOXC6* mRNA in patients with advanced GC (AGC) were found to be significantly higher than those in patients with early-stage GC (EGC) [29]. Another report suggested that the combination of three biomarkers (collagen type VI alpha 3 chain (*COL6A3*), serpin family H member 1 (*SERPINH1*), and pleckstrin homology and RhoGEF domain containing G1 (*PLEKHG1*)) yielded an elevated AUC of 0.907. A higher *COL6A3* level was also significantly correlated with lymph node metastasis and poor prognosis in GC patients, while high levels of SER*PINH1* and *PLEKHG1* mRNA expression were correlated with lower overall survival (OS) in GC patients [30]. CircRNAs may also be an auxiliary diagnostic biomarker of GC [16]. In a combined 11 studies, which included 12 types of circRNAs (11 in tissues and 5 in plasma), all were downregulated. The combined diagnostic OR (DOR) and AUC with 95% CI were 8.778 (6.108, 12.614) and 0.81 (0.78, 0.84) respectively [31]. Another type of RNA with diagnostic value for GC detection include long non-coding RNAs (lncRNAs). The ROC curve showed that the AUC of serum lncRNA HCP5 detected by qRT-PCR was 0.818 (95% CI 0.757–0.880, *p* < 0.001, 80% sensitivity, and 70% specificity), while the three combined diagnoses (HCP5, CEA, and CA199) provided the highest AUC of 0.870 (95% CI 0.819–0.921, *p* < 0.001) in distinguishing between GC and healthy donors reaching 81% sensitivity and 79% specificity [32]. In addition, it was found that LINC00941 was associated with tumor depth and distant metastasis in GC as it could discriminate GC samples from normal samples (AUC = 0.7911, 95% CI 0.7264–0.8559, *p* < 0.0001) and M1 samples from M0 samples (AUC = 0.6809, 95% CI 0.5852–0.7766, *p* = 0.0031) [33].

### 4.2. Limitations, Future Studies, and Advantage of the Markers Used in This Study

The abovementioned studies suggest salivary RNAs as potential biomarkers for the diagnosis of GC, but emphasize the need for validation in intended use populations. However, no study of adequate sample size for independent validation has been performed to date. There remains an unmet need to develop a noninvasive biomarker assay for identifying patients with GC from a high-risk population. Thus, the major advantage of the markers used in the current study is their noninvasive nature, which is important for population-based screening in at-risk populations.

## 5. Conclusions

We aimed to develop universal biomarkers for GC that could be applicable to all individuals regardless of their ethnic origin. Although we were unable to ”validate” the prior model developed based on a Korean cohort [7], we were able to demonstrate that our markers had diagnostic utility above and beyond demographic factors alone. Additional studies are needed to evaluate the diagnostic utility of our models in different ethnic populations, such as a Korean cohort in a U.S. population. More importantly, our study emphasizes the importance of population-specific biomarker development and validation for salivary exRNA biomarker for GC detection.

## Figures and Tables

**Figure 1 cancers-14-03632-f001:**
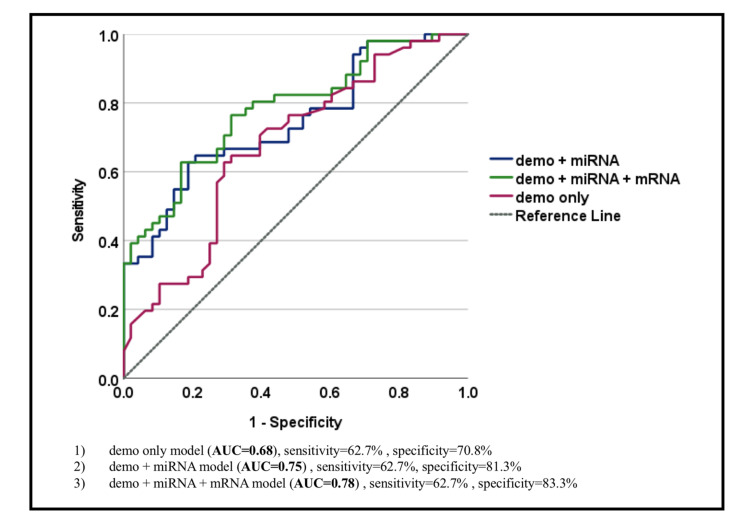
Performance of 3 different models: (**1**) A new model with only demographic characteristics (AUC = 0.68, sensitivity = 62.7%, and specificity = 70.8%); (**2**) a new model with demographic characteristics and miRNA biomarkers for gastric cancer (AUC = 0.75, sensitivity = 62.7%, and specificity = 81.3%); (**3**) a new model with demographic characteristics, miRNA and mRNA biomarkers for gastric cancer (AUC = 0.78, sensitivity = 62.7, and specificity = 83.3%).

**Figure 2 cancers-14-03632-f002:**
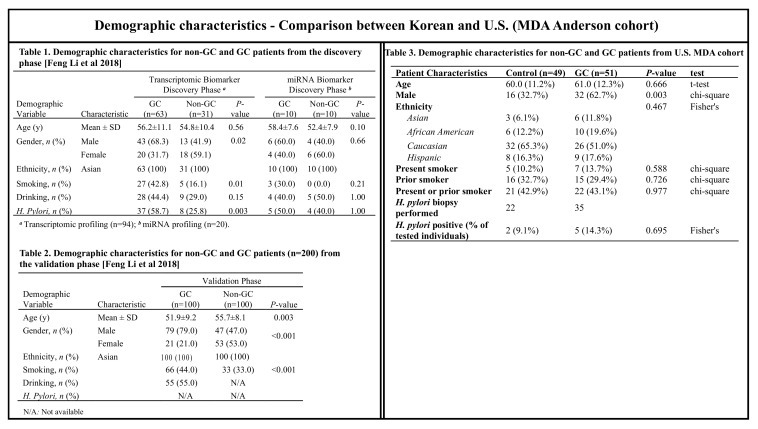
Comparison of demographic characteristics between Korean and U.S. study groups. The demographic characteristics of the Korean cohort used in our previous study [7] (Table 1 for the discovery phase and Table 2 for the validation phase) were compared with the demographic characteristics of the U.S. cohort used in this study (Table 3). (Tables 1 and 2 adapted from Li et al. [7]. Reprinted with permission of the Oxford University Press, Copyright © 2022 American Association of Clinical Chemistry. Li et al., Discovery and Validation of Salivary Extracellular RNA Biomarkers for Noninvasive Detection of Gastric Cancer. *Clin. Chem.* 2018, *64*, 1513–1521. https://doi.org/10.1373/clinchem.2018.290569).

**Table 1 cancers-14-03632-t001:** Patients’ characteristics for non-GC and GC patients from the MD Anderson Cancer Center.

Patient Characteristics	Control (*n* = 49)	GC (*n* = 51)	*p*-Value	Test
**Age**	60.0 (11.2%)	61.0 (12.3%)	0.666	*t*-Test
**Male**	16 (32.7%)	32 (62.7%)	0.003	Chi-square
**Ethnicity**			0.467	Fisher’s
Asian	3 (6.1%)	6 (11.8%)		
Black, non-Hispanic	6 (12.2%)	10 (19.6%)		
Caucasian	32 (65.3%)	26 (51.0%)		
Hispanic	8 (16.3%)	9 (17.6%)		
**Present smoker**	5 (10.2%)	7 (13.7%)	0.588	Chi-square
**Prior smoker**	16 (32.7%)	15 (29.4%)	0.726	Chi-square
**Present or prior smoker**	21 (42.9%)	22 (43.1%)	0.977	Chi-square
***H. pylori* biopsy performed**	22	35	--	--
***H. pylori* positive (% of tested individuals)**	2 (9.1%)	5 (14.3%)	0.695	Fisher’s

**Table 2 cancers-14-03632-t002:** Models: (A) Demographic model for gastric cancer and (B) demographic model with two miRNA biomarkers for gastric cancer.

**A. Demographic Model for Gastric Cancer**		
**Terms**	**Odds Ratio (95% CI)**	***p*-Value**
Age	0.99 (0.96–1.04)	0.945
Male	3.74 (1.57–8.92)	0.003
Present or prior smoker	0.75 (0.31–1.79)	0.514
**B. Demographic Model with Two miRNA Biomarkers for Gastric Cancer**		
**Terms**	**Odds Ratio (95% CI)**	***p*-Value**
Age	0.99 (0.95–1.03)	0.683
Male	5.42 (2.03–14.48)	0.001
Ever Smoker	0.82 (0.33–2.07)	0.680
dCT*miR140_U6*	2.56 (1.37–4.79)	0.003
dCT*miR301_U6*	0.36 (0.19–0.68)	0.002

**Table 3 cancers-14-03632-t003:** Performance of the panel (demographic features + miRNAs + mRNAs) according to the different stages of GC (I–IV).

vs. Control	Overall	Stage I/II	Stage III/IV	Stage IV
**Demographic features only**	0.68 (0.57–0.78)	0.67 (0.51–0.84)	0.67 (0.56–0.79)	0.68 (0.55–0.80)
**Demographic features + miRNAs**	0.75 (0.65–0.84)	0.80 (0.63–0.96)	0.72 (0.61–0.83)	0.70 (0.58–0.83)
**Demographic features + miRNAs + mRNAs**	0.78 (0.69–0.87)	0.85 (0.72–0.99)	0.75 (0.64–0.85)	0.74 (0.63–0.86)

**Table 4 cancers-14-03632-t004:** Demographic characteristics with miRNA and mRNA biomarkers for gastric cancer.

Demographic Features + 2 miRNA Biomarkers for GC + 3 mRNA Biomarkers for GC		
**Terms**	**OR (95% CI)**	***p*-Value**
Age	0.99 (0.95–1.03)	0.544
Male	5.42 (2.03–14.48)	0.001
Ever Smoker	0.82 (0.33–2.07)	0.421
dCT*miR-140*_*U6*	2.56 (1.37–4.79)	0.007
dCT*miR-301a*_*U6*	0.36 (0.19–0.68)	0.002
dCT*PPL*_*ACTB*	0.88 (0.66–1.18)	0.406
dCT*SEMA4B*_*ACTB*	0.90 (0.66–1.23)	0.497
*dCTSPINK7_ACTB*	1.23 (0.94–1.60)	0.132

## Data Availability

All data can be found in the text.

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
