# Peer review of "Performance of Salivary Extracellular RNA Biomarker Panels for Gastric Cancer Differs between Distinct Populations"

_cancers, 2022, doi:10.3390/cancers14153632_

Round 1
Reviewer 1 Report
In the original research article “Ethnic Diversity in extracellular RNA biomarkers for detection of gastric cancer” the authors present the results from a potential model to detect gastric cancer. The model developed by the group uses extracellular RNA (microRNA and mRNA) levels as predictors for whether an individual has gastric cancer. The model has been previously tested and validated in Korea with only Asian patients. This study demonstrates its applicability to a more diverse population in the United States. The introduction is adequate in providing reason for the study. Unfortunately, the methods, results, discussion, and conclusion are not substantial enough for publication. The following provides a more detailed explanation of what needs to be addressed:
Major Concerns:
- Authors do not mention, nor provide, patient consent information regarding this study. Additionally, authors should include the IRB number when mentioning its approval.
- The article heavily relies on the group’s previous publication “Discovery and Validation of Salivary Extracellular RNA Biomarkers for Noninvasive Detection of Gastric Cancer” to explain their model, methodology, and reasoning. For a proper understanding of their investigations, it is not possible for the current article under consideration to stand alone.
- In addition to the previous point, this study lacks novelty considering analyses were similar to the previous publication and there was ample data to do further investigations. For example, a key question that should be answered is whether or not there are differences in biomarker levels across the different ethnicities.
- The title of the article is misleading as there are no real investigations as to how extracellular RNA relates to ethnic diversity.
- This study states to have ethnic diversity and thus, makes the claim that their model can be applied across multiple ethnicities. Since their Hispanic and Black populations remain relatively low, I am not convinced there is enough to power to make these claims. More patients and patient diversity should be included.
- Authors do not investigate nor address any differences that could be different with the studies being conducted in two different parts of the world and how this could affect their model.
- I am curious as to why the authors used U6 for their reference gene as there are multiple articles suggesting its invalidity: https://pubmed.ncbi.nlm.nih.gov/25450382/
- The results are not clearly presented and difficult to follow. For example, it took the reader until the discussion to determine that low expression of microRNA and mRNA markers was indicative of a diagnosis of gastric cancer. With this, it is unclear in this article specifically how these varying levels could be used to diagnose gastric cancer.
- Figure 2 is difficult to read due to size and impossible to follow with the legend provided.
- In line 265 of the discussion, Table 4 is improperly referenced.
- In line 287, it states more microRNAs were tested for then described in the methods.
- Authors in the discussion should have compared their studies to other cancers using extracellular RNA as a cancer detection tool. Overall, the discussion seems brief with inadequate investigation of other research being conducted in this field of study.
- The final sentence of the conclusion seems contradictory to the title and claims made in this study.
Minor Concerns:
- In the methods section, authors should include the formula used for quantifying qPCR and they should also include a subsection on their statistics. I do not think for this type of article, the statistics should be included in the results section.
- The labeling of the methods section is not properly done. Each subsection has the same number.
- There are several abbreviations that appear before being written out.
- Human genes need to be italicized.
Reviewer 2 Report
Interesting work on the identification of salivary exRNA biomarkers for GC detection in the US. However, I have a few comments to make about the manuscript:
- Line: 59-60: gastric cancer incidence and mortality data are from 2017. Update these data to estimates for 2021 or 2022.
- Line 68-69: gastric cancer is the fourth in incidence worldwide, but does it have a low incidence in the US? What position does it occupy with respect to other types of cancer?
- Figure 2: I suggest improving the quality of the image and increasing the font size of the tables that appear at the top (table 1 and table 2).
- I suggest carrying out a greater bibliographic search and increasing the number of references used.
Reviewer 3 Report
It’s a nice study but requires further exploration. The sample size is less (n=50) and it’s difficult to conclude any outcome. It’s premature to say biomarker at this stage. “250 ml of cell-free saliva” is it not too much volume? As the sample size is less, and its model based prediction, statistical part should be checked properly.
Round 2
Reviewer 1 Report
Thank you to the authors for providing a thorough response to the first reviewer’s report. Extensive explanations were included and substantial changes were made to address initial concerns. Unfortunately, after much consideration, the manuscript falls short in adequacy for publication in Cancers. The following are the main reasons for this conclusion:
1. Although the authors explain the novelty of studying a new cohort, the manuscript still heavily relies on their previous publication studying the same RNA biomarkers in a Korean population. The only difference is the investigation of the ability to use a select demographic categories (for which their selection is unclear) to distinguish GC from non-GC patients in a population of individuals where GC is less common. The group would greatly benefit from studying more patients and from different regions of the country.
2. Collectively, the entire manuscript is lacking enough data/analyses for publication in Cancers. Understandably, the collection of samples and performing qPCR can be an extensive process; however, there was ample opportunity for the authors to perform other analysis as it relates to miRNA expression (e.g., disease severity, response to therapy, event-free survival, etc.).
3. The authors themselves mention the need for validation of their model, of which, such validation would make the manuscript stronger and increase its potential for publication in Cancers.
The authors should be applauded for finding these miRNA biomarkers and it will be exciting to see their future findings should they conduct further analyses as well as examine other cohorts. If the authors do not wish to conduct further experimentation and is of interest, this work might be suitable for publication in another MDPI journal such as Diagnostics.
Reviewer 3 Report
May be accepted